# Metabolomic Biomarker Candidates for Skeletal Muscle Loss in the Collagen-Induced Arthritis (CIA) Model

**DOI:** 10.3390/jpm11090837

**Published:** 2021-08-26

**Authors:** Paulo V. G. Alabarse, Jordana M. S. Silva, Rafaela C. E. Santo, Marianne S. Oliveira, Andrelise S. Almeida, Mayara S. de Oliveira, Mônica L. Immig, Eduarda C. Freitas, Vivian O. N. Teixeira, Camilla L. Bathurst, Claiton V. Brenol, Lidiane I. Filippin, Stephen P. Young, Priscila S. Lora, Ricardo M. Xavier

**Affiliations:** 1VA San Diego Healthcare System, Veteran Medical Research Foundation, San Diego, CA 92161, USA; 2Hospital de Clínicas de Porto Alegre, Serviço de Reumatologia, Laboratório de Doenças Autoimunes, Porto Alegre 90035-903, Brazil; jo-msouza@hotmail.com (J.M.S.S.); rcsanto@hcpa.edu.br (R.C.E.S.); m.schraderdeoliveira@gmail.com (M.S.O.); andrelisealmeida.biomed@gmail.com (A.S.A.); mayarasouzaoliveira2@gmail.com (M.S.d.O.); moniimmig@hotmail.com (M.L.I.); eduarda.freitas.hcpa@gmail.com (E.C.F.); viviont@gmail.com (V.O.N.T.); claiton.brenol@gmail.com (C.V.B.); rxavier10@gmail.com (R.M.X.); 3Queen Elizabeth Hospital and National Institute for Health Research, Birmingham Biomedical Research Centre, University of Birmingham, Birmingham B15 2TT, UK; c.l.bathurst@bham.ac.uk (C.L.B.); s.p.young@bham.ac.uk (S.P.Y.); 4Universidade La Salle, Canoas 92010-000, Brazil; lidifillipin@gmail.com; 5Universidade do Vale do Rio dos Sinos, São Leopoldo 93022-750, Brazil; priscilaslora@gmail.com

**Keywords:** rheumatoid arthritis, precision medicine, NMR, CIA, metabolomics, cachexia, biomarkers

## Abstract

There is no consensus for diagnosis or treatment of RA muscle loss. We aimed to investigate metabolites in arthritic mice urine as biomarkers of muscle loss. DBA1/J mice comprised collagen-induced arthritis (CIA) and control (CO) groups. Urine samples were collected at 0, 18, 35, 45, 55, and 65 days of disease and subjected to nuclear magnetic resonance spectroscopy. Metabolites were identified using Chenomx and Birmingham Metabolite libraries. The statistical model used principal component analysis, partial least-squares discriminant analysis, and partial least-squares regression analysis. Linear regression and Fisher’s exact test via the MetaboAnalyst website were performed (VIP-score). Nearly 100 identified metabolites had CIA vs. CO and disease time-dependent differences (*p* < 0.05). Twenty-eight metabolites were muscle-associated: carnosine (VIPs 2.8 × 10^2^) and succinyl acetone (VIPs 1.0 × 10) showed high importance in CIA vs. CO models at day 65; CIA pair analysis showed histidine (VIPs 1.2 × 10^2^) days 55 vs. 65, histamine (VIPs 1.1 × 10^2^) days 55 vs. 65, and L-methionine (VIPs 1.1 × 10^2^) days 0 vs. 18. Carnosine was fatigue- (0.039) related, creatine was food intake- (−0.177) and body weight- (−0.039) related, and both metabolites were clinical score- (0.093; 0.050) and paw edema- (0.125; 0.026) related. Therefore, muscle metabolic alterations were detected in arthritic mice urine, enabling further validation in RA patient’s urine, targeting prognosis, diagnosis, and monitoring of RA-mediated muscle loss.

## 1. Introduction

Rheumatoid arthritis (RA) is an autoimmune inflammatory disease characterized by symmetric polyarthritis and systemic involvement [1]. It affects about 1% of people under 35 years old and more than 2% of adults over 60 years in the United States, and its prevalence has been reported worldwide [2]. While the joints are the main target of the disease, there are many extra-articular manifestations, such as body composition changes, which are strongly associated with long-term disability and premature mortality [3]. In fact, significant loss of muscle mass is widely described in the literature as one of the major metabolism-related changes seen in RA [4].

As a common and important complication of RA, the loss of muscle mass has been associated with the intensity of inflammation and the severity of disease [5]. Low muscle quantity, quality, and strength are the main features of sarcopenia, a muscle disease (muscle failure) associated with several other conditions [6]. In RA, the presence of sarcopenia has been reported in around 28–37% of patients [7,8,9]. Moreover, the RA muscle loss may occur with or without loss of fat mass, resulting in limited or absent changes in body mass index (BMI), characterizing rheumatoid cachexia (RC) [10]. The RC definition contrasts with classic cachexia, which is characterized by severe weight loss, mainly loss of muscle mass, and increased protein catabolism due to an underlying illness [11]. In a systematic review with meta-analysis, the prevalence of RC was 15–32%, while classical cachexia was absent [12].

The precise assessment of skeletal muscle status and its changes over time is the main problem to diagnose and treat body composition changes. Currently, there are several ways to estimate muscle loss, as can be listed: anthropometrics (e.g., BMI), bioelectrical impedance analysis, imaging techniques (computer tomography, nuclear resonance imaging, dual X-ray absorptiometry, ultrasound), biochemical analysis (assessment of total potassium), urinary creatinine amount, and different tests to quantify muscle function. However, they either lack diagnostic value (sensitivity, specificity) or are high in cost [13,14]. In this context, the search for novel biomarkers related to muscle loss is important to provide the attending physician with better ways to predict development, stage, and progression of muscle involvement during the routine follow-up of RA patients.

Metabolomics is a systems biology technique that uses a ‘top-down’ approach in which data is gathered at the systemic level [15]. The metabolome is defined as the complete set of metabolites from 100 to 1000 Da present in each biological system and currently represents a strong new tool for research [16,17]. Metabolic footprinting comprises the analysis of the extracellular metabolites produced by the organism and can be used to discriminate differences between health and diseased states and to investigate promising biomarkers for diseases [18].

As a comprehensive and sensible technique, the metabolomics analysis is reliable and reproducible and has been used in RA in a comprehensive way, e.g., for disease characterization and treatment response prediction [15,16,18,19]. Considering that the collagen-induced arthritis (CIA) model shares many similarities with RA, the purpose of this study was to analyze the urinary metabolomic profile associated with loss of muscle mass in CIA animals over time. 

## 2. Materials and Methods

This study demonstrates the in vivo data of disease induction, animal follow-up and urine collection, as well as metabolomic analysis. Detailed in vivo experimentation methods were published and are described in the Appendix A in the Journal of Cachexia, Sarcopenia and Muscle Wasting. As all CIA mice presented muscle loss following the disease induction, in the present study the urine metabolomic profile of these mice was analyzed [20].

Animals: Male DBA/1J mice from 8 to 12 weeks of age were used. The mice were reared alone at 20 °C, with 12-h light–dark cycle, and free access to food and water. The animals were randomly divided into two experimental groups: (i) healthy animals (CO, *n* = 11) and (ii) collagen-induced arthritic animals (CIA, *n* = 13). Animals were followed up for 65 days and all measurements were done prior to the arthritis induction and thereafter at days 18, 25, 35, 45, 55, and 65. All experiments were performed in accordance with the Guiding Principles for Research Involving Animals and in accordance with the Ethics Committee of Research and Postgraduate Group of the Clinical Hospital of Porto Alegre (Hospital de Clínicas de Porto Alegre—HCPA; number. 14-0297).

Disease induction: Arthritis was induced with bovine type II collagen (CII, Chondrex, Inc., Woodinville, USA; 2 mg/mL) dissolved in 0.1 M acetic acid at 4 °C for 12 h, and Complete Freund’s Adjuvant (CFA; Sigma, St. Louis, MO, USA; 2 mg/mL) containing inactivated *Mycobacterium tuberculosis*. Fifty microliters of emulsion (CII + CFA) were intradermally injected at the base of the tail to induce arthritis; it was set as the day zero in this experiment. Eighteen days after the first injection, the animals received a reinforcement of CII emulsified with incomplete Freund’s adjuvant (IFA—without *Mycobacterium tuberculosis*) in another site of the tail (booster injection). During the procedures, mice were anesthetized with isoflurane 10% (Abbott) and 90% of oxygen [21]. Healthy control mice were manipulated and anesthetized; however, no injection was made. Animals were euthanized 65 days after the first injection.

Animal models follow up: The following experiments were performed and the results, which had already been published by the group, were used for correlation with the metabolites in the present study. At days 18, 25, 35, 45, 55, and 65, clinical severity score, hind paw edema, body weight, food intake, free exploratory locomotion, grip strength, and endurance exercise performance of the mice were evaluated. At the end of the experimental period, animals were euthanized, and the tibialis anterior and gastrocnemius muscles were dissected and weighed to confirm cachexia development. The tibiotarsal joint was collected to confirm the development of arthritis by histological analysis with Hematoxylin-Eosin staining [20].

Urine collection: Animals were confined individually in metabolic cages (Tecniplast S.p.A., Buguggiate, Italy) for 6 h to collect urine before the arthritis induction and thereafter at days 18, 25, 35, 45, 55, and 65. Urine was filtered in 0.22-µm filters and stored in −80 °C until further nuclear magnetic resonance spectroscopy (NMR) analysis.

NMR sample preparation and analysis: Urine samples were thawed on ice and centrifuged for 5 min, at 13,000× *g* at 4 °C. Supernatants were collected and 4× NMR buffer (final concentration: 100 mM of phosphate, 2 mM of difluorotrimethylsilanylphosphonic acid (DFTMP, Manchester Organics, Manchester, UK), 10% D2O, 0.1% azide, 0.5 mM of 4,4-dimethyl-4-silapentane-1-sulfonic acid) was added and mixed. Then samples were centrifuged for 5 min, at 13,000× *g* at 4 °C, and supernatants were transferred into glass champagne vials, capped and frozen at −80 °C until the NMR analysis. Samples were thawed and transferred to 1.7 mm NMR tubes (Bruker Biospin, Coventry, UK) using an Anachem Autosampler. One-dimensional 1H spectra were acquired at 300 K using a standard 1D-1H-Nuclear Overhauser Effect spectroscopy (NOESY) pulse sequence with water saturation using pre-sat in a Bruker AVANCE II 600 MHz NMR spectrometer (Bruker Corp., Billerica, USA) equipped with a 1.7-mm cryoprobe. Spectral width was set to 12 ppm, and the scans were repeated 128 times. Sample’s series were loaded into 96-tube racks and held at 6 °C in the SampleJet sample handling device until processed. Two-dimensional 1H J-resolved (JRES) spectra were also acquired to aid metabolite identification.

NMR spectra metabolic identification and pathway analysis: Chenomx software (Chenomx Inc., Edmonton, Canada) and the associated metabolites library were used to identify the compounds in 1D NMR spectra. The 2D-Jres NMR spectra were submitted to Birmingham Metabolome Library (BML) to identify and quantify their intensities. MetaboAnalyst website (http://www.metaboanalyst.ca/, 1 July 2021) [22] was used to elucidate which pathways the most relevant metabolites were related to [16,19,20].

Metabolomic network: Cytoscape analysis (the Cytoscape 3.3; cytoscape.org) was used to visualize and to analyze the biological networks between metabolic profiles and clinical parameters of the animal model follow-up. Data were imported in Metascape, a plugin of Cytoscape, and subjected to pathway analysis. All the metabolic pathways generated were further subjected to enrichment and topological analysis. Only the pathways having significant *p*-value were selected and further subjected to identification of biological processes associated. Compounds, reactions, enzymes, genes, pathways, and the relationships among them provided the initial framework for the metabolomic data analysis [23].

Statistics: Sample size was based on the previous research of our group in which the main outcome was muscle atrophy accessed by myofiber area in CIA model [24]. Considering an alpha of 5% and the statistical power of 90% to detect differences in tibialis anterior myofiber area, a sample size of seven animals per group was required. Due to the possibility of death of about 10% of the animals during the disease induction procedure, or during the disease course, the final sample size was 11 mice for the CO group and 13 mice for the CIA group. 1D NMR spectra were binned, normalized, glog-transformed, and subjected to principal component analysis (PCA) and partial least-squares discriminant analysis (PLSDA) to group data, followed by a venetian blind test to obtain specificity and sensibility of the model. PCA and PLSDA were also used to analyze BML results. Partial least-squares regression analysis (PLSR) was performed using the BML metabolic profile with each clinical data, as previously described [18]. Briefly, PLSR is a regression method that identifies which metabolites can predict a continuous variable (in vivo analysis was considered a continuous variable). This analysis yields R2, a measure of the cross-validated goodness-of-fit of the linear regression, and each metabolite contribution to the model, while permutation testing (multiple analyses using random data subsets) was used to assess the significance of this prediction. Pathway analysis was performed applying the Fisher’s exact test by the MetaboAnalyst website and to reach pathway impact [25,26,27]. Statistical significance was always set for a *p*-value under 0.05. The PLSDA model of the variable of importance in the projection score critical limit was set for a value higher than 1 × 10°, which means a *p*-value under 0.05. Metabolite list picking was manually done through KEGG (www.genome.jp/kegg, 1 July 2021) [28], PUBCHEM (pubchem.ncbi.nlm.nih.gov) [29], and HMDB (www.hmdb.ca, 1 July 2021) [30] websites in which it was checked if there was a direct relationship with muscle metabolism.

## 3. Results

In vivo experimental: CIA animals had significantly higher arthritis scores and hind paw edema volumes than CO. In addition, from histopathology, the disease was confirmed in all CIA animals. The CIA group showed no weight loss or decreased appetite. However, the normalized weights (i.e., divided by the body weight of the animal) for visceral and brown fat were reduced in CIA animals (54% and 39%, respectively) and ankle joint normalized weights and spleen normalized weights were increased in the CIA group when compared to CO (18% and 40%, respectively). CIA animals showed lower free exploratory locomotion, endurance exercise performance total time, and grip strength compared to the CO group. The dissected tibialis anterior and gastrocnemius muscles weighed less in the CIA than in the CO group (25% and 24%, respectively) and sarcoplasmic ratios were also smaller in CIA when compared with CO (23% and 22% less sarcoplasmic ratio, respectively). Thus, the myofiber diameter reduction was 45% in TA and 41% in GA [20]. After the confirmation of muscle loss in CIA mice, the non-targeted metabolome was acquired to identify all metabolites present in the urine of these animals during the development of arthritis [20]. To search for potential biomarkers of muscle loss in cachectic mice, urine samples from these same cohort of animals were collected for metabolic profiling analysis at seven time points (0, 18, 25, 35, 45, 55 and 65 days) during the 65 days of disease. Profile comparisons between CIA and CO mice, and between each time point within the CIA group, were performed aiming to characterize metabolic profiles of the diseased animals and their changes over time.

PCA and PLSDA models: Firstly, we developed an unsupervised PCA model using all the NMR processed spectra and it was followed by PLSDA supervised analysis to group them into clusters for group homogeneity (Figure 1A,B). The main component and latent variable of each sample was plotted with mean ± standard deviation of the mean (Figure 1C,D).

To reduce complexity, we performed a PCA model followed by PLSDA at each time point comparing CIA and CO (paired analysis; Figure 2 and Figure 3, PCA and PLSDA, respectively). In addition, to elucidate the disease course within CIA animals, we conducted paired analyses between the different time points (Figure 4 and Figure 5) in the CIA group. As can be seen, these PLSDA models could segregate CO from CIA groups as well as CIA groups in a time-dependent manner. Lastly, all the previously mentioned models from PLSDA were submitted to a venetian blind test to provide sensitivity and specificity values to be considered, as shown in Table 1. As observed by PCA, method samples have been homogeneously grouped with almost no samples outside the confidence interval (CI; Figure 2 and Figure 4). Based on the venetian blind test done after PLSDA, we decided to maintain all samples that could be considered outliers because those samples added more specificity and sensibility to the model than when removed (Figure 3 and Figure 5). Taken together, the built models were robust enough to provide a long list of statistical significance (VIP score higher than 1 × 10°) metabolites that were analyzed for muscle loss process.

Metabolic profiling: Considering the whole set of urine samples, almost 400 metabolites were identified, and time-dependent significant differences between CIA vs. CO, and CIA vs. CIA were found (Figure 6). From the list of all metabolites with statistically significant differences (data not shown), we speculate that these differences were related to the arthritis disease and to the processes involved in its development, which may have effects in all organs and tissues, in addition to the muscle loss process.

From all metabolites, 28 metabolites were associated with pathways related to muscle tissues, including muscle catabolic and anabolic processes based on information from KEGG, PUBCHEM, and HMDB websites (Table 2). The following metabolites showed high importance in the models: carnosine (VIP score 2.8 × 10^2^) from CIA vs. CO at 65 days; histidine (VIP score 1.2 × 10^2^) from CIA pair analysis days 55 vs. 65; histamine (VIP score 1.1 × 10^2^) from CIA pair analysis days 55 vs. 65; L-methionine (VIP score 1.1 × 10^2^) from CIA pair analysis days 0 vs. 18; and succinyl acetone (VIP score 1.0 × 10^2^) from CIA vs. CO at 65 days. As expected, given their importance in muscle protein structure, there were also several amino acids with high statistical differences between the compared groups, i.e., high variable importance in the projection score.

Biomarker time-dependent identification:

After assessing the statistically significant differences in metabolites levels, these were then linked to pathways known to occur within muscle tissues. These metabolites were displayed in a timeline, according to the analysis results of different time points (Figure 7a). Before 45 days of established muscle loss presence, we identified the 3-methylhistidine, 4-aminobutyric acid, acetylcholine, arginine, aspartate, carnosine, creatine, creatinine, glutamine, histamine, histidine, isoleucine, leucine, l-methionine, lysine, myo-inositol, *N,N*-dimethylglycine, *N*-acetyl alanine, *N*-acetylmethionine, pantothenate, phenylalanine, phosphocholine, phosphocreatine, pyridoxine, sarcosine, succinyl acetone, and thiamine. After 45 days, from the previous list, aspartate, phenylalanine, and thiamine were excluded for the purpose of follow up. On the other hand, urocanate was included. The metabolites elected also are shown in a heatmap (Figure 7b) as well as a connection network with the weight each connection has (Figure 7c).

Several metabolites were related to the same 18 pathways that occur within muscle tissues, and 11 pathways were related to amino acid metabolism (Figure 8). Furthermore, the four most frequent pathways were: histidine metabolism (e.g., metabolites: 3-methylhistidine, carnosine, histamine, histidine, and urocanate); arginine and proline metabolism (arginine, 4-aminobutyrate, creatine, and phosphocreatine); glycine, serine, and threonine metabolism (*N*,*N*-dimethylglycine, sarcosine, phosphocholine); and creatine phosphate metabolism (creatine, creatinine, phosphocreatine). Some of these pathways also presented statistical differences between several time points, as shown in Table 3.

Briefly, histidine metabolism was the most frequent pathway altered during disease development compared to controls. Concerning metabolite frequency within the same metabolic pathway, histidine had the highest impact of 0.60 comparing the days 55 and 65 in CIA. Arginine and proline metabolisms also appeared frequently, with an impact of up to 0.28; valine, leucine, and isoleucine metabolisms appeared in almost every time point comparing CIA and CO, and had an impact of up to 0.33.

Regression analysis: Finally, we performed a regression analysis between the clinical data and metabolites from urine samples, and identified them by BML (Table 4). Although the regressions models generated with these data presented statistically significant differences, the highest R2 found in the models were weak, i.e., from 0.04 to 0.3. PLSR for grip strength showed that with 24 metabolites (including *N*-dimethylglycine, thiamine, and arginine) the model reached a maximum contribution R2 of 0.30 (*p* < 0.01); endurance exercise performance showed that with 50 metabolites (including *N*-acetyl alanine, sarcosine, *N*-acetylmethionine, phosphocholine, carnosine, and glutamine) the model reached a maximum contribution R2 of 0.11 (*p* < 0.17); free locomotion required 40 metabolites (including thiamine, *N*-acetyl alanine, sarcosine, arginine, and carnosine) to reach a maximum contribution R2 of 0.04 (*p* < 0.03); clinical score required 26 metabolites (including creatinine, methionine, carnosine, leucine, and glutamine) to reach a maximum contribution R2 of 0.05 (*p* < 0.03); hind paw edema showed that with 12 metabolites (including carnosine, leucine, creatinine, and 4-aminobutyrate) the model reached a maximum contribution R2 of 0.084 (*p* < 0.08); food intake required 15 metabolites (including *N*-acetyl alanine, creatinine, and 3-methylhistidine) to reach a maximum contribution R2 of 0.30 (*p* < 0.08); and body weight required 10 metabolites (including creatinine) to reach maximum contribution R2 of 0.06 (*p* < 0.01). Table 4 shows the R2 contribution value from the PLSR model for the statistically significant correlations between the 22 muscle-related metabolites and the clinical data. The metabolites R2 not related to muscle metabolism are not shown. Additionally, the 3-methylhistidine, 4-aminobutyric acid, acetylcholine, arginine, aspartate, carnosine, creatine, creatinine, glutamine, histamine, histidine, isoleucine, leucine, l-methionine, lysine, myo-inositol, *N,N*-dimethylglycine, *N*-acetyl alanine, *N*-acetylmethionine, pantothenate, phenylalanine, phosphocholine, phosphocreatine, pyridoxine, sarcosine, succinyl acetone, and thiamine were associated with muscle loss processes (Figure 8).

## 4. Discussion

Several metabolites and pathways related to muscle catabolism and anabolism were altered in urine of CIA mice compared to controls and relatively to the stage of disease development. Namely, the metabolites 3-methylhistamine, 4-aminobutyric acid, acetylcholine, arginine, carnosine, creatine, creatinine, glutamine, histamine, histidine, isoleucine, leucine, l-methionine, lysine, myo-inositol, *N,N*-dimethylglycine, *N*-acetyl alanine, *N*-acetylmethionine, pantothenate, phosphocholine, phosphocreatine, pyridoxine, sarcosine, succinyl acetone, and urocanate, which are involved in important signaling pathways for muscle tissues, showed more impact following the metabolomic analysis. Although muscle loss is an important feature in RA and in experimental arthritis, the number of studies is limited, and the correlation of muscle impairment in human and animal models is unclear [31]. Thus, the altered profiles found in this study may be valuable for the clinical assessment of RA muscle loss. 

As observed in the Venn diagram (Figure 6), the analysis brought a list of 54 unique identified metabolites by the BML, 265 uniquely identified by ChenomX-NMR, and 154 common metabolites found in both automatic (BML) and manual (ChenomX—NMR) assessments. With all these metabolites, we could access an initial list of nearly 100 metabolites after the first steps of statistical analysis (PCA, PLSDA, and Venetian blind test). Afterwards, using the pathway analysis, and confirming the metabolites list in the KEGG, PUBCHEM, and HMDB databases, and performing the PLS-regression, we finally could reduce this list down to 27 metabolites previously mentioned. All these steps favor building a robust statistical set to reach the most muscle-loss linked list of metabolites in this research.

As previously demonstrated by our group, DBA/1J mice with CIA present approximately 30% decrease in myofiber cross-sectional area after 45 days of disease induction and are considered cachectic at the sixty-fifth day [20,24]. Continuing the investigation about arthritis muscle loss, we performed metabolomics analysis to explore potential biomarkers in the urine of these cachectic mice. The use of animal models has the advantage of circumventing the great heterogeneity implicit in human studies, increasing the power to prospect useful biomarkers. Previous metabolomics studies in RA have addressed profiles for disease activity, drug exposure, and prognosis in early disease, demonstrating the feasibility of this approach [15,17,18,19,32]. A systematic review compiled metabolomics studies, which used serum, urine, and synovial fluid for the analysis, and designated the following metabolites as important in RA: glucose, lactic acid, citric acid, leucine, methionine, isoleucine, valine, phenylalanine, threonine, serine, proline, glutamate, histidine, alanine, cholesterol, glycerol, and ribose [33]. Furthermore, studies reported that muscle features can also be measured using a metabolomics approach. Reduced amine metabolites have been found in plasma of CIA mice indicating that disordered amine response may be linked to the muscle wasting and to the increased resting energy expenditure mediated by RA [34,35]. The analysis of urine collected from CIA rats treated with tetrandrine described 23 potential biomarkers associated with CIA, which were mainly linked to metabolism of energy, amino acid, lipid, and gut microbe [36]. These findings in RA are similar to ours, demonstrating that metabolomics may be an interesting approach not only for disease prognosis and monitoring, but also to assess the muscle loss related to disease pathogenesis.

The histidine metabolism was the pathway most frequently altered in CIA mice. Furthermore, all metabolites related to this pathway (3-methylhistidine, carnosine, histamine, histidine, and urocanate) had major statistical significance in the PLSDA model analysis between CIA and CO groups. Histidine was described as one of the amino acids’ end products, highly excreted by RA patients, and as a discriminatory metabolite for RA treatment prediction [19]. Compared to controls, histidine serum levels were found lower in RA patients at baseline, and significantly higher 3 months after the treatment with TNF-α inhibitors [37]. Additionally, both RA and experimental arthritis showed better outcomes following the supplementation with carnosine [38,39]. Concerning the regression analysis, carnosine was related to fatigue, free locomotion, clinical score, and hind paw edema best models, while 3-methylhistidine was related to food intake. Accordingly, both metabolite 3-methyl-histidine and carnosine were identified as markers of muscle loss, and oral administration of histidine was able to improve grip strength and walking speed in chronic kidney disease patients [40,41]. In addition, lower levels of carnosine dipeptidase 1were associated with cancer cachexia, compared with weight stable patients [42]. Therefore, as histidine pathway has a strong relationship with muscle deficits, the metabolites of this pathway may potentially be validated as biomarkers for muscle loss in RA.

Arginine and proline metabolisms were also affected by CIA, and specifically the metabolites arginine, 4-aminobutyrate, creatine, and phosphocreatine had high statistical significance in PLSDA models between CIA and CO groups. Previously, metabolomics profiles of RA synovial fluid showed that arginine and proline pathways are downregulated with the disease, while serum levels of L-proline were increased in RA patients compared to controls [43,44]. Additionally, the metabolite 4-aminobutyrate, also known as GABA, was suggested to have influence in RA due to a regulatory role on inflammation [45]. The influence of the semi-essential amino acid, arginine, on muscle tissues, is related to the production of creatine, urea, and nitric oxide (NO), and to the synthesis of new proteins [46]. Through its vasodilatory function, NO may play a role in nutrient delivery to the muscle, as well as in healing or fibrosis fate [46,47,48,49,50]. Regarding muscle function, arginine was related to grip strength and free locomotion in the best regression models. Accordingly, it has been demonstrated that serum arginine was suitable to differentiate RA patients into classes of physical disability [51]. Thus, we can suggest a role of arginine and proline metabolisms in RA muscle loss process.

As creatine phosphate metabolism is deeply important for muscle function it was not surprising that metabolites related to this pathway were observed several times in our data. Creatinine levels are strongly associated with muscle energetic metabolism and wasting and are increased in individuals with muscle loss [52]. Creatinine is the degradation product of creatine when it is properly phosphorylated when used in the muscle energy processes. Although creatine supplementation in RA patients increases muscle mass, it does not change strength or physical function [53]. This metabolite has appeared in the best regression models for clinical score, hind paw edema, food intake, and body weight. As formerly mentioned, this pathway was used to monitor muscle mass change, being a strong candidate for validation as a biomarker also in RA. 

Glycine, serine, and threonine compose metabolic pathways related to amino acids, and the metabolites *N*,*N*-dimethylglycine, sarcosine, and phosphocholine were frequent in PLSDA models. *N*,*N*-dimethylglycine has been positively correlated with fat-free mass in chronic obstructive pulmonary disease patients [40]. Otherwise, increased phosphocholine levels may have indirect effects on muscle loss, since murine cancer cells from tumors, which are capable of inducing cachexia, presented increased phosphocholine levels compared to tumors unable to induce cachexia [54]. Regarding muscle function, *N*,*N*-dimethylglycine contributed to the best model of grip strength, which agrees with the potential role of glycine related to fatigue previously described [55]. In RA, serum glycine was suitable to differentiate RA patients into classes of physical disability [51]. Therefore, directly, or indirectly, this pathway also characterizes a good biomarker candidate for muscle loss in RA.

In addition to pathways discussed before, the influence of other amino acids within muscle metabolism have been reported. Glutamine, in which muscle levels are known to be an important supply of this amino acid to the body, appeared in both endurance exercise performance and clinical score best regression models [56]. The availability of glutamine is reduced by conditions such as inactivity, and its low muscle concentration is associated with decreased protein synthesis in acute disease states [56,57]. Regarding methionine, which appeared in the best regression model for clinical score, it is the primary amino acid required to initiate protein synthesis, as well as acute phase proteins synthesis [58]. Leucine contributed to the best PLSR model for edema; this amino acid is suggested as a stimulator of protein synthesis, with potential to reduce muscle loss in mice with adenocarcinoma after supplementation [59,60]. Notwithstanding, metabolites not related to the amino acid metabolism, such as B vitamin complex components, also have anabolic roles in skeletal muscle tissue [61]. The best regression models for grip strength and free locomotion were related to thiamine.

Some statistically significant metabolites found in PLSDA in our model were not extensively discussed because of the paucity of literature regarding muscle metabolism process and/or related to chronic inflammatory process. These metabolites are listed here as follow: urocanate (histidine metabolism), succinyl acetone (tyrosine metabolite), *N*-acetylmethionine (methionine derivative), *N*-acetyl alanine (substrate from a variety of cellular reactions), acetylcholine (a neurotransmitter at neuromuscular junction), sarcosine (a derivative of glycine), and aspartate (a non-essential proteinogenic amino acid). 

The present study has some limitations: lack of kidney function data or proteinuria to confirm any interference to the metabolic profile. Another limitation is that BML is an automatic detection system that cannot be controlled, resulting in a lack of detection of some metabolites, and leading to a less precise analysis; however, performing the NMR analysis functioned as a complementary analysis, and confirmed the automated BML findings. Despite this, our findings should not be neglected.

## 5. Conclusions

In conclusion, several of the metabolites found in the urine of CIA mice with muscle loss were related to muscle metabolism. Obtaining these results in an animal model highlights the possibility of relevant findings following the analysis of urine metabolites in RA patients. Of note, our group is already performing research that aims to investigate the urine metabolic profiles of RA subjects, and relate them to RC and sarcopenia phenotypes, in addition to analyzing the association of these conditions with the disease activity in patients (data not yet published). The addressed issue is important, and if confirmed in human studies, may be important for early individuation of subjects at risk for muscle wasting in RA. The identification and quantification of metabolites in RA patients would be useful to predict and to monitor the muscle involvement of the disease, since the alterations in pathways may differ among patients with diverse genetic and phenotypic backgrounds. Additionally, muscle loss is not limited to RA, but occurs in other several chronic inflammatory diseases, and in the aging process as sarcopenia. Thus, the metabolites could be validated as biomarkers for these diseases as well.

## Figures and Tables

**Figure 1 jpm-11-00837-f001:**
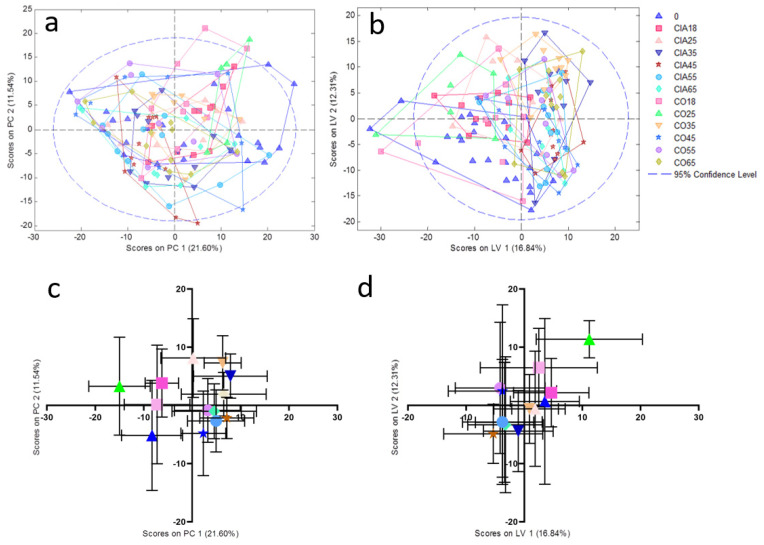
PCA (**a**) and PLSDA (**b**) analysis of CIA and CO groups with all-time points. (**c**) and (**d**) represent the plotted data in mean ± standard deviation of the mean from PCA and PLSDA analysis, respectively. 0: all animals before the beginning of the experiment (blue triangle); CIA18: CIA group 18 days after disease induction (pink square); CIA25: CIA group 25 days after disease induction (light pink triangle); CIA35: CIA group 35 days after disease induction (dark blue inverted triangle); CIA45: CIA group 45 days after disease induction (red star); CIA55: CIA group 55 days after disease induction (light blue circle); CIA65: CIA group 65 days after disease induction (light blue diamond); CO18: CO group 18 days after the experiment beginning of the (light pink square); CO25: CO group 25 days after the beginning of the experiment (green triangle); CO35: CO group 35 days after the beginning of the experiment (orange inverted triangle); CO45: CO group 45 days after the beginning of the experiment (blue star); CO 55: CO group 55 days after the beginning of the experiment (purple circle); CO65: CO group 65 days after the beginning of the experiment (yellow diamond). PC: principal component; LV: latent variable.

**Figure 2 jpm-11-00837-f002:**
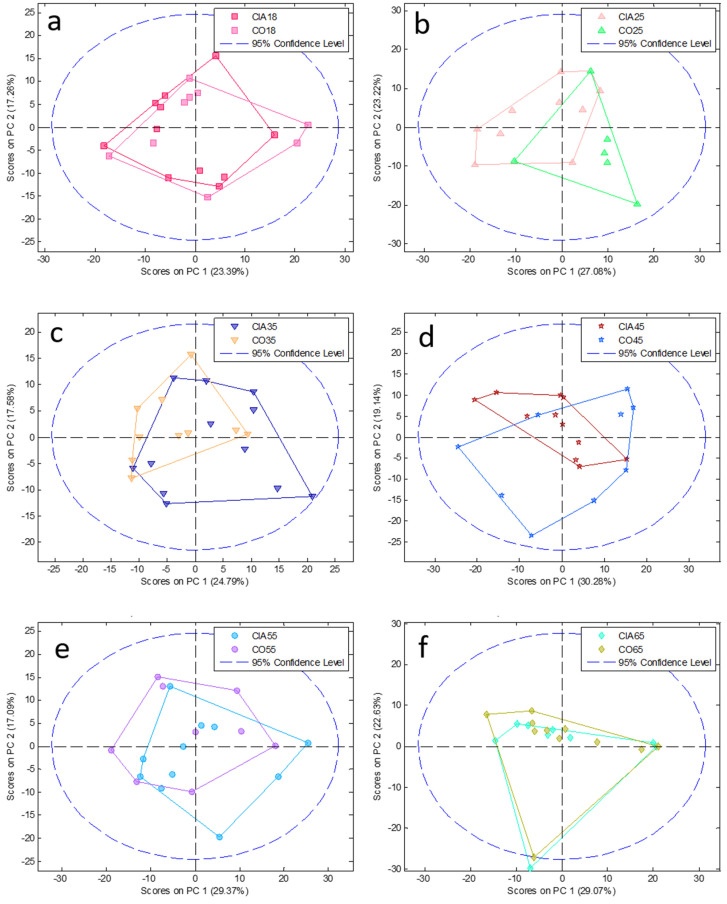
PCA analysis at each time point between CIA and CO (pair analysis). Graphics from (**a**–**f**) show comparison among 18, 25, 35, 45, 55, and 65 days, respectively. CIA18: CIA group 18 days after induction (pink square); CIA25: CIA group 25 days after disease induction (light pink triangle); CIA35: CIA group 35 days after disease induction (dark blue inverted triangle); CIA45: CIA group 45 days after disease induction (red star); CIA55: CIA group 55 days after disease induction (light blue circle); CIA65: CIA group 65 days after disease induction (light blue diamond); CO18: CO group 18 days after the beginning of the experiment (light pink square); CO25: CO group 25 days after the beginning of the experiment (green triangle); CO35: CO group 35 days after the beginning of the experiment (orange inverted triangle); CO45: CO group 45 days after the beginning of the experiment (blue star); CO55: CO group 55 days after the beginning of the experiment (purple circle); CO65: CO group 65 days after the beginning of the experiment (yellow diamond). PC: principal component.

**Figure 3 jpm-11-00837-f003:**
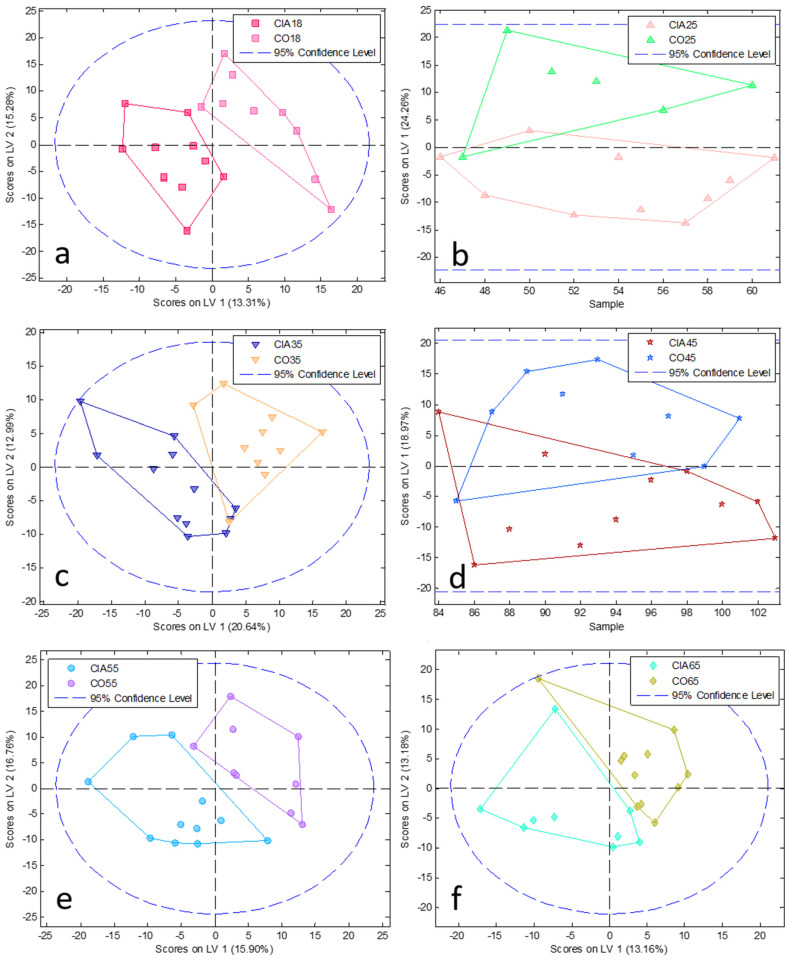
PLSDA analysis at each time point between CIA and CO (pair analysis). Graphics from (**a**–**f**) show comparison among 18, 25, 35, 45, 55, and 65 days, respectively. CIA18: CIA group 18 days after disease induction (pink square); CIA25: CIA group 25 days after disease induction (light pink triangle); CIA35: CIA group 35 days after disease induction (dark blue inverted triangle); CIA45: CIA group 45 days after disease induction (red star); CIA55: CIA group 55 days after disease induction (light blue circle); CIA65: CIA group 65 days after disease induction (light blue diamond); CO18: CO group 18 days after the beginning of the experiment (light pink square); CO25: CO group 25 days after the beginning of the experiment (green triangle); CO35: CO group 35 days after the beginning of the experiment (orange inverted triangle); CO45: CO group 45 days after the beginning of the experiment (blue star); CO55: CO group 55 days after the beginning of the experiment (purple circle); CO65: CO group 65 days after the beginning of the experiment (yellow diamond). LV: latent variable.

**Figure 4 jpm-11-00837-f004:**
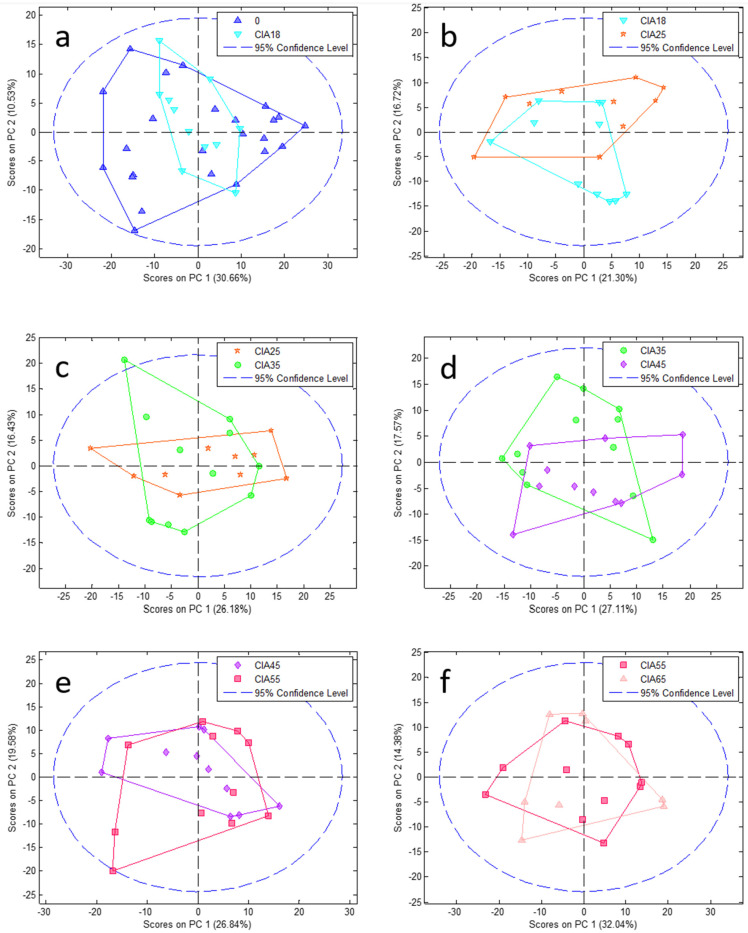
PCA pair analysis of CIA groups from one-time point to the next one. Graphics from (**a**–**f**) show comparison among 0 vs. CIA 18, CIA 18 vs. CIA 25, CIA 25 vs. CIA 35, CIA 35 vs. CIA 45, CIA 45 vs. CIA 55, and CIA 55 vs. CIA 65 days, respectively. 0: all animals before the experiment beginning (blue triangle); CIA18: CIA group 18 days after disease induction (light blue inverted triangle); CIA25: CIA group 25 days after disease induction (red star); CIA35: CIA group 35 days after disease induction (green circle); CIA45: CIA group 45 days after disease induction (purple diamond); CIA55: CIA group 55 days after disease induction (pink square); CIA65: CIA group 65 days after disease induction (light pink triangle). PC: principal component.

**Figure 5 jpm-11-00837-f005:**
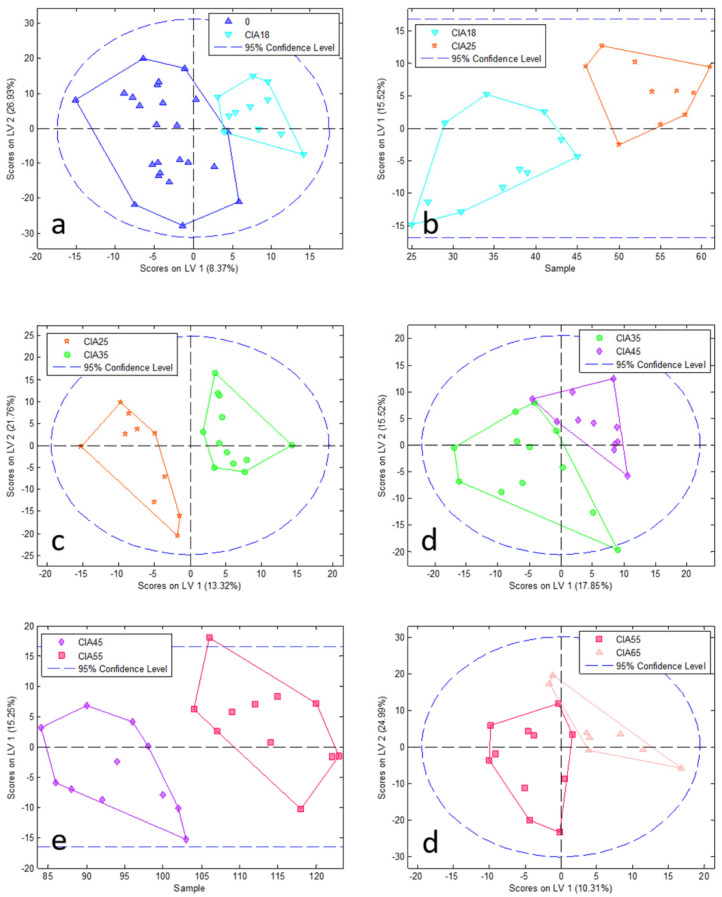
PLSDA pair analysis of CIA groups from one-time point to the next one. Graphics from (**a**–**f**) show comparison among 0 vs. CIA 18, CIA 18 vs. CIA 25, CIA 25 vs. CIA 35, CIA 35 vs. CIA 45, CIA 45 vs. CIA 55, and CIA 55 vs. CIA 65 days, respectively. 0: all animals before the experiment beginning (blue triangle); CIA18: CIA group 18 days after disease induction (light blue inverted triangle); CIA25: CIA group 25 days after disease induction (red star); CIA35: CIA group 35 days after disease induction (green circle); CIA45: CIA group 45 days after disease induction (purple diamond); CIA55: CIA group 55 days after disease induction (pink square); CIA65: CIA group 65 days after disease induction (light pink triangle). LV: latent variable.

**Figure 6 jpm-11-00837-f006:**
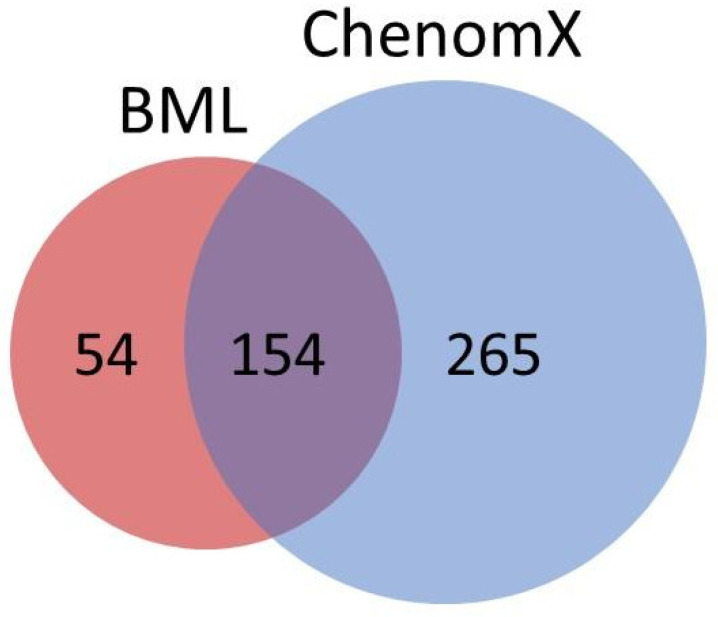
Venn diagram showing source of identified metabolites from BML (red) and ChenomX (blue).

**Figure 7 jpm-11-00837-f007:**
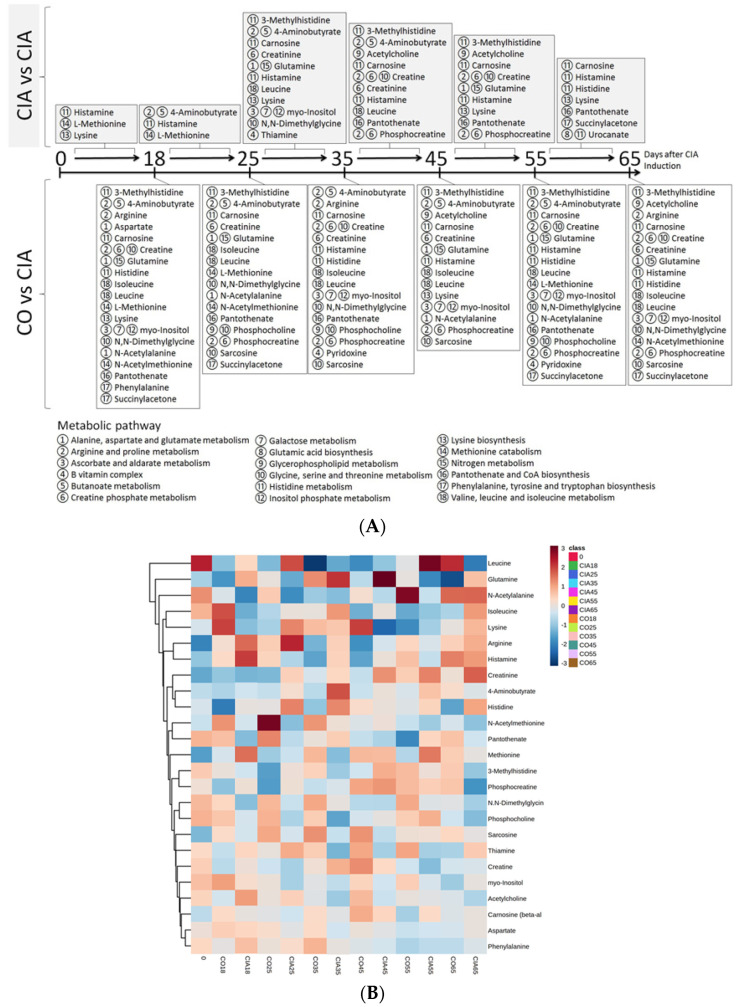
(**A**) The timeline according to the analysis results of different time points. (**B**) Heatmap of the selected metabolite from BML related to muscle loss. (**C**) Network of the selected metabolite (blue diamonds) with intensity of connections (red: higher intensity; blue: lower intensity) from BML. Pathway analysis.

**Figure 8 jpm-11-00837-f008:**
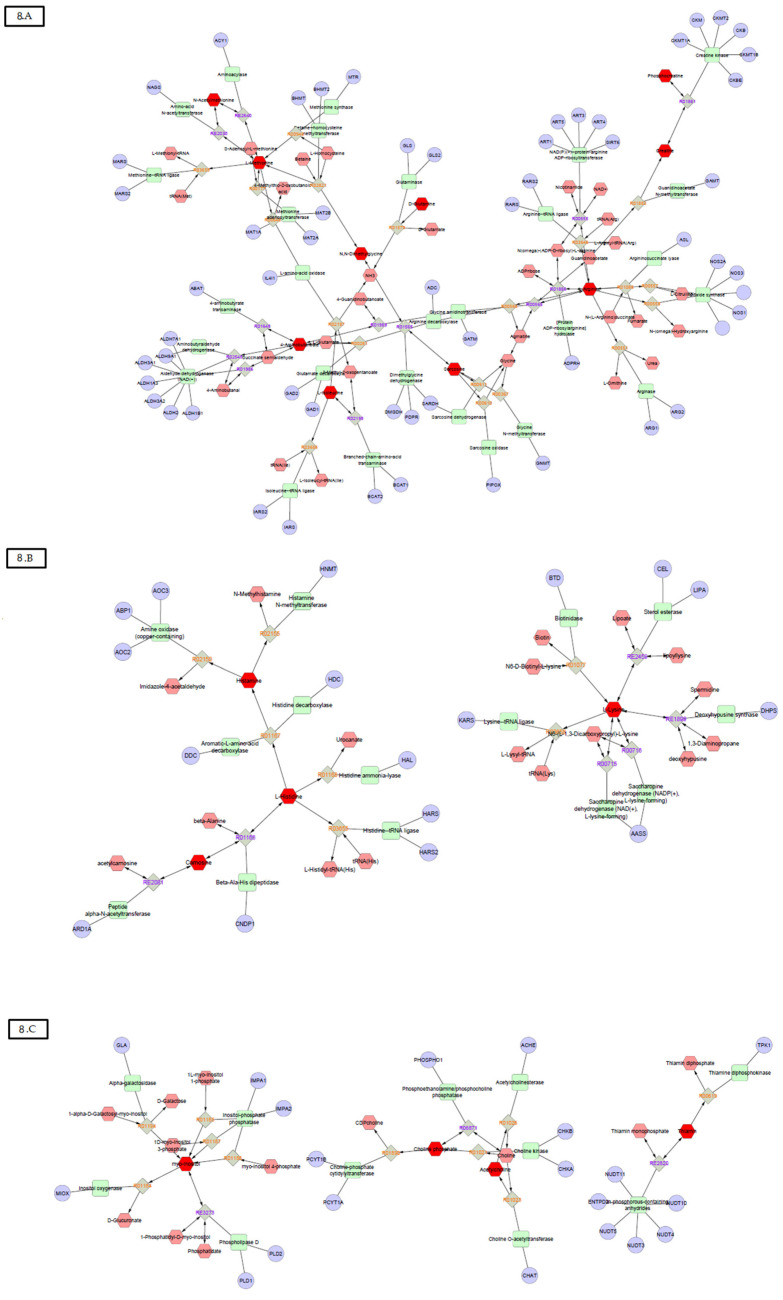
Pathways that are related to amino acid metabolism.

**Table 1 jpm-11-00837-t001:** Specificity and sensibility of NMR data after PLSDA model built obtained by the venetian blind test.

Group	Calibration		Cross Validated	
CO vs. CIA Same Time Point	Sensitivity	Specificity	Sensitivity	Specificity
CO 18 and CIA 18	1.00	1.00	0.92	0.67
CO 25 and CIA 25	0.90	0.83	0.90	0.67
CO 35 and CIA 35	1.00	0.90	0.75	0.70
CO 45 and CIA 45	0.82	0.78	0.73	0.56
CO 55 and CIA 55	1.00	1.00	0.55	0.33
CO 65 and CIA 65	1	1	0.444	0.636
Pair analysis				
0 and CIA 18	1	1	0.792	0.727
CIA 18 and CIA 25	0.727	0.9	0.636	0.8
CIA 25 and CIA 35	1	1	0.9	0.917
CIA 35 and CIA 45	1	1	0.667	0.727
CIA 45 and CIA 55	0.727	0.727	0.273	0.091
CIA 55 and CIA 65	1	1	0.545	0.25
ALL GROUPS				
T 0	0.88	0.80	0.79	0.77
CIA 18	0.83	0.66	0.75	0.66
CIA 25	0.70	0.57	0.60	0.62
CIA 35	0.83	0.62	0.83	0.64
CIA 45	1.00	0.56	0.82	0.55
CIA 55	0.909	0.439	0.636	0.485
CIA 65	0.667	0.403	0.667	0.425
CO 18	0.444	0.784	0.333	0.776
CO 25	0.833	0.788	0.667	0.781
CO 35	0.9	0.752	0.8	0.789
CO 45	0.667	0.664	0.556	0.657
CO 55	0.778	0.381	0.667	0.373
CO 65	0.909	0.523	0.636	0.53
CIA ONLY				
CIA 18	0.92	0.93	0.67	0.89
CIA 25	0.60	0.73	0.50	0.75
CIA 35	0.92	0.76	0.75	0.70
CIA 45	0.91	0.52	0.73	0.56
CIA 55	0.82	0.54	0.64	0.52
CIA 65	0.667	0.55	0.56	0.54

**Table 2 jpm-11-00837-t002:** Metabolites associated with muscle catabolic and anabolic pathways.

Metabolite	Statistics Comparison Origin (NMR or BML) ^a^	Variable Importance in Projection Score
3-Methylhistidine	CIA pair analysis 45 × 55 (NMR)	1.0 × 10^1^
	CIA vs. CO at 18, 25, 45, 55, and 65 (BML)	6.6 × 10°, 4.8 × 10°, 2.7 × 10°, 1.9 × 10°, 2.0 × 10°
	CIA pair analysis 25 × 35, 35 × 45 (BML)	2.0 × 10°, 1.3 × 10°
4-Aminobutyrate	CIA vs. CO at 18, 25, 35, 45, and 55 days (BML)	1.3 × 10°, 3.6 × 10^1^, 1.7 × 10^1^, 1.4 × 10°, 1.5 × 10°
	CIA pair analysis 18 × 25, 25 × 35, 35 × 45 (BML)	1.8 × 10°, 3.5 × 10°, 4.9 × 10°
Acetylcholine	CIA vs. CO at 45, and 65 days (BML)	1.6 × 10°, 1.4 × 10°
	CIA pair analysis 35 × 45, and 45 × 55 (NMR)	4.9 × 10^1^, 3.8 × 10^1^
Arginine	CIA vs. CO at 18, 35, and 65 days (BML)	1.9 × 10°, 2.4 × 10°, 1.8 × 10°
Aspartate	CIA vs. CO at 18 days (BML)	2.0 × 10°
Carnosine	CIA vs. CO at 65 days (NMR)	2.8 × 10^2^
	CIA pair analysis 25 × 35 (NMR)	3.6 × 10°
	CIA vs. CO at 18, 25, 35, 45, 55, and 65 days (BML)	2.1 × 10°, 1.7 × 10^1^, 4.6 × 10^1^, 1.2 × 10^1^, 2.9 × 10^1^, 5.0 × 10°
	CIA pair analysis 25 × 35, 35 × 45, 45 × 55, 55 × 65 (BML)	5.9 × 10°, 1.6 × 10^1^, 3.1 × 10°, 4.4 × 10^1^
Creatine	CIA pair analysis 35 × 45, 45 × 55 (NMR)	2.3 × 10°, 8.4 × 10^1^
	CIA vs. CO at 18, 35, 55, and 65 days (BML)	1.8 × 10°, 1.4 × 10°, 3.3 × 10°, 1.4 × 10°
Creatinine	CIA vs. CO at 45 days (NMR)	3.4 × 10^1^
	CIA pair analysis 35 × 45 (NMR)	3.1 × 10^1^
	CIA vs. CO at 25, 35, 45, and 65 days (BML)	3.3 × 10°, 9.2 × 10°, 1.6 × 10°, 4.0 × 10°,
	CIA pair analysis 25 × 35	3.7 × 10°
Glutamine	CIA vs. CO at 18, 25, 45, 55, and 65 (BML)	3.1 × 10°, 1.8 × 10°, 3.2 × 10°, 3.4 × 10°, 2.4 × 10°
	CIA pair analysis 25 × 35, 45 × 55 (BML)	5.7 × 10°, 1.2 × 10°
Histamine	CIA vs. CO at 35, 45, 55, and 65 days (NMR)	2.8 × 10^1^, 3.7 × 10^1^, 6.2 × 10^1^, 5.2 × 10^1^
	CIA pair analysis 0 × 18, 18 × 25, 25 × 35, 35 × 45, 45 × 55, 55 × 65 (NMR)	8.7 × 10°, 8.5 × 10°, 4.9 × 10°, 1.3 × 10^1^, 1.9 × 10^1^, 1.1 × 10^2^
	CIA vs. CO at 35, 45, and 65 days (BML)	3.9 × 10°, 3.7 × 10°, 4.9 × 10°
	CIA pair analysis 25 × 35 (BML)	4.2 × 10°
Histidine	CIA vs. CO at 18 days (NMR)	2.6 × 10^1^
	CIA pair analysis 55 × 65 (NMR)	1.2 × 10^2^
	CIA vs. CO at 35, 55, and 65 days (BML)	2.5 × 10°, 1.2 × 10°, 2.5 × 10°
Isoleucine	CIA vs. CO at 45 days (NMR)	2.1 × 10^1^
	CIA vs. CO at 18, 25, 35, 45, and 65 days (BML)	5.3 × 10°, 1.1 × 10°, 1.5 × 10°, 1.7 × 10°, 1.7 × 10°,
Leucine	CIA vs. CO at 18, 25, 35, 45, 55, and 65 days (BML)	1.0 × 10°, 2.7 × 10°, 3.5 × 10°, 1.1 × 10°, 1.5 × 10°, 2.2 × 10°
	CIA pair analysis 25 × 35, 35 × 45 (BML)	1.8 × 10°, 1.6 × 10°
L-Methionine	CIA vs. CO at 18 days (NMR)	5.4 × 10^1^
	CIA pair analysis 0 × 18, 18 × 25 (NMR)	1.1 × 10^2^, 2.1 × 10°
	CIA vs. CO at 25, and 55 days (BML)	1.2 × 10°, 5.1 × 10°
Lysine	CIA pair analysis 0 × 18, 25 × 35, 45 × 55, 55 × 65 (NMR)	2.8 × 10^1^, 5.3 × 10°, 3.1 × 10°, 6.1 × 10°, 2.5 × 10°
	CIA vs. CO at 18, and 45 days (BML)	5.4 × 10°, 3.2 × 10°
myo-Inositol	CIA vs. CO at 18, 35, 45, 55, and 65 days (BML)	1.7 × 10°, 4.9 × 10°, 1.1 × 10°, 1.6 × 10°, 1.2 × 10°
	CIA pair analysis 25 × 35 (BML)	2.1 × 10^1^
*N*,*N*-Dimethylglycine	CIA vs. CO at 18, 25, 55, and 65 days (BML)	3.5 × 10°, 2.4 × 10°, 2.6 × 10°, 1.3 × 10°
	CIA pair analysis 25 × 35 (BML)	4.1 × 10°
*N*-Acetylalanine	CIA vs. CO at 18, 25, 45, and 55 days (BML)	1.5 × 10°, 2.8 × 10°, 1.7 × 10°, 5.4 × 10°
*N*-Acetylmethionine	CIA vs. CO at 18, 25, and 65 days (BML)	3.1 × 10°, 8.7 × 10°, 3.4 × 10°
Pantothenate	CIA vs. CO at 35 days (NMR)	2.1 × 10^1^
	CIA pair analysis 35 × 45, 45 × 55, 55 × 65 (NMR)	1.4 × 10^1^, 3.3 × 10^1^, 8.8 × 10°
	CIA vs. CO at 18, 25, 35, and 55 days (BML)	2.7 × 10°, 1.8 × 10°, 1.0 × 10°, 1.8 × 10°
Phenylalanine	CIA vs. CO at 18 days (BML)	1.8 × 10°
Phosphocholine	CIA vs. CO at 25, 35, and 55 days (BML)	4.3 × 10°, 1.6 × 10°, 1.7 × 10°
Phosphocreatine	CIA vs. CO at 45 days (NMR)	3.4 × 10^1^
	CIA pair analysis 35 × 45, 45 × 55 (NMR)	3.1 × 10^1^, 8.4 × 10^1^
	CIA vs. CO 25, 35, 55, and 65 days (BML)	1.9 × 10°, 2.9 × 10°, 6.2 × 10°, 3.3 × 10°
Pyridoxine	CIA vs. CO at 35, and 55 days (NMR)	4.6 × 10^1^, 5.2 × 10^1^
Sarcosine	CIA vs. CO at 25, 35, 45, and 65 days (BML)	1.0 × 10°, 3.3 × 10°, 2.3 × 10°, 1.9 × 10°
Succinylacetone	CIA vs. CO at 18, 25, 55, and 65 days (NMR)	7.0 × 10^1^, 4.7 × 10^1^, 6.8 × 10^1^, 3.7 × 10^1^, 1.0 × 10^2^
	CIA pair analysis 55 × 65 (NMR)	5.2 × 10^1^
Thiamine	CIA pair analysis 25 × 35 (BML)	2.8 × 10°
Urocanate	CIA pair analysis 55 × 65 (NMR)	5.0 × 10°

^a^, NMR: 1D nuclear magnetic resonance; BML: 2D J-res nuclear magnetic resonance identified by the Birmingham Metabolome Library.

**Table 3 jpm-11-00837-t003:** Metabolic pathway analysis of metabolite ranking data from PLSDA models comparing CIA vs. CO at each time-point and CIA pair analysis at each time. Data expressed as “% of metabolic pathway impact (*p*-value)”.

BML	Alanine, Aspartate, and Glutamate Metabolism	Arginine and Proline Metabolism	Ascorbate and Aldarate Metabolism	B Vitamin Complex	Butanoate Metabolism	Glycerophospholipid Metabolism	Glycine, Serine, and Threonine Metabolism	Histidine Metabolism	Lysine Metabolism	Nitrogen Metabolism	Pantothenate and CoA Biosynthesis	Valine, Leucine, and Isoleucine Degradation
CIA pair analysis 0 × 18								22 (0.002)	0 (0.066)			
CIA pair analysis 18 × 25		14 (0.040)			3 (0.326)			22 (0.027)				
CIA pair analysis 25 × 35	26 (0.070)	28 (*p* < 0.001)	0 (0.154)		3 (0.337)		3 (0.109)	22 (0.029)		0 (0.154)		33 (0.185)
CIA pair analysis 35 × 45		22 (0.007)			3 (0.337)	11 (0.103)	2 (0.109)	0 (0.244)				33 (0.185
CIA pair analysis 45 × 55	15 (0.074)	21 (0.001)				11 (0.110)	0 (0.453)	22 (0.032)	0 (0.074)	0 (0.159)		
CIA pair analysis 55 ×65									0 (0.002)			
NMR												
CIA pair analysis 0 × 18								22 (0.166)	0 (0.030)			
CIA pair analysis 18 × 25								22 (0.130)				
CIA pair analysis 25 × 35							0 (0.284)	22 (0.010)	0 (0.023)			2 (0.336)
CIA pair analysis 35 × 45		1 (0.487)				0 (0.364)	0 (0.374)	22 (0.202)			2 (0.202)	2 (0.437)
CIA pair analysis 45 × 55		1 (0.503)				0 (0.378)		22 (0.210)	0 (0.001)		2 (0.021)	
CIA pair analysis 55 × 65								61 (0.001)	0 (0.001)		2 (0.184)	
CIA vs. CO at 18 days	6 (0.113)				0 (0.104)		0 (0.144)	24 (0.072)	0 (0.020)	0 (0.044)		
CIA vs. CO at 25 days	0 (0.113)				0 (0.104)		0 (0.009)					0 (0.053)
CIA vs. CO at 35 days					0 (0.010)		0 (0.199)	22 (0.101)			2 (0.101)	0 (0.075)
CIA vs. CO at 45 days	0 (0.227)	0 (0.378)			0 (0.021)		0 (0.284)	22 (0.148)	0 (0.042)			0 (0.111)
CIA vs. CO at 55 days	0 (0.157)			8 (0.062)	0 (0.145)	4 (0.193)	0 (0.199)	22 (0.101)				0 (0.075)
CIA vs. CO at 65 days							0 (0.144)	22 (0.002)				2 (0.174)

**Table 4 jpm-11-00837-t004:** Regression analysis between clinical data and metabolic profile found in urine and identified by BML. Metabolites shown are solely the ones related to muscle loss process. Value expresses the metabolite contribution to the model.

	Strength	Fatigue	Locomotion	Score	Edema	Intake	Weight
3-Methylhistidine	0.044	−0.033	−0.020	−0.020	−0.051	0.130	−0.048
4-Aminobutyate	−0.073	−0.071	−0.016	0.056	0.097	0.022	0.001
Acetylcholine	0.063	0.015	0.015	−0.024	−0.036	0.068	−0.018
Arginine	0.083	−0.004	0.031	−0.093	−0.035	0.004	0.029
Carnosine	−0.039	0.039	0.031	0.093	0.125	−0.051	−0.026
Creatine	−0.063	−0.014	−0.051	0.050	0.026	−0.177	−0.039
Creatinine	−0.056	−0.061	−0.062	0.153	0.107	0.142	0.058
Glutamine	−0.019	0.024	0.003	0.058	0.066	0.066	0.024
Histamine	−0.029	−0.035	−0.007	0.001	0.036	−0.004	0.019
Histidine	−0.048	−0.040	0.022	0.033	0.059	0.036	−0.050
Isoleucine	−0.046	−0.028	−0.018	0.052	0.049	0.000	−0.030
L-Methionine	−0.082	0.003	−0.006	0.109	0.048	−0.020	0.024
Leucine	−0.101	−0.069	−0.028	0.083	0.121	0.041	−0.022
Lysine	0.020	−0.006	−0.020	−0.033	−0.070	−0.180	−0.080
myo-Inositol	−0.011	−0.026	−0.044	0.039	0.008	0.050	−0.012
*N*-Acetylalanine	0.028	0.093	0.063	0.031	−0.008	0.214	0.032
*N*-Acetylmethionine	0.014	0.056	0.021	−0.060	−0.001	0.035	0.014
*N*,*N*-Dimethylglycine	0.107	−0.022	−0.014	−0.044	−0.070	−0.006	−0.020
Phosphocholine	0.016	0.041	0.005	0.022	−0.025	0.022	0.000
Phosphocreatine	0.033	0.023	−0.010	0.037	0.060	0.000	−0.007
Sarcosine	0.037	0.066	0.053	−0.017	−0.023	0.083	0.028
Thiamine	0.084	0.012	0.126	−0.047	−0.156	0.025	0.000

Values express the metabolite contribution to the model.

## Data Availability

Not applicable.

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
