# Peer review of "Metabolomic Biomarker Candidates for Skeletal Muscle Loss in the Collagen-Induced Arthritis (CIA) Model"

_jpm, 2021, doi:10.3390/jpm11090837_

Round 1
Reviewer 1 Report
I have no further comments on this article.
Author Response
Thank you for reading our manuscript another time, and for the careful and constructive reviews. We are sorry to hear that one of the reviewers could not find the letter, then, we have carefully added the answers of the first group of questions from the first letter at the end of this file. In addition, we have made every effort to respond to all the comments and state the changes performed on the manuscript. We believe we have addressed all the suggestions recommended by the reviewers and we are confident that the new version of the manuscript is greatly improved. A revised version of the manuscript is now submitted for your appraisal. The responses to each point raised by the reviewers are detailed below. We look forward to hearing from you regarding our submission. And we would be glad to respond to any further questions and comments that you may have. Please see the attachment.

Reviewer 2 Report
I thank the authors for their work in addressing my previous queries. However, it was difficult to follow the revision process, since I could not see a clear response of the authors addressing these. Usually resubmission of revised manuscripts is accompanied by a point-by-point response letter from the authors replying to each of the issues raised by reviewers. I believe this could help the authors addressing several of my previously raised questions, since for many of them I do not see any changes introduced to the manuscript, and in this way it is hard to understand the reasoning of the authors for this. In line with this, I believe the following aspects still need to be clarified before the manuscript is acceptable for publication:
1. The differential performance of PCA and PLSDA analysis in segregating CO and CIA groups was not explained. A discussion on the validity and performance of both methods for the analysis of this data should be included in the discussion section.
2. The authors did not seem to have addressed what do they mean with their sentence “all the previously mentioned models from PLSDA provided sensitivity and specificity values to be considered, as shown in Table 1” (page 5, line 239). It is still not clear what are the “sensitivity and specificity values” the authors are referring to.
3. The authors included a Venn diagram showing the distribution of metabolites across the BML an NMR datasets. However, it is still missing a discussion about what in these datasets could be triggering the identification of different major pathways implicated in a specific disease timepoint (e.g., it is not clear why histidine pathway is identified with a 61% contribution to disease progression from day 55 to day 65 in the NMR dataset, but completely absent from the BML dataset).
4. Taking into the account the temporal variation of the metabolites depicted in Figure 7.B it is still not clear the choice of all the metabolites proposed as disease biomarkers up to day 45 of disease development. This should be further clarified.
Minor point: the manuscript should be revised for English grammar and syntax, particularly the new parts of text introduced during revision.
Author Response
Thank you for reading our manuscript another time, and for the careful and constructive reviews. We are sorry to hear that one of the reviewers could not find the letter, then, we have carefully added the answers of the first group of questions from the first letter at the end of this file. In addition, we have made every effort to respond to all the comments and state the changes performed on the manuscript. We believe we have addressed all the suggestions recommended by the reviewers and we are confident that the new version of the manuscript is greatly improved. A revised version of the manuscript is now submitted for your appraisal.
The responses to each point raised by the reviewers are detailed below. We look forward to hearing from you regarding our submission. And we would be glad to respond to any further questions and comments that you may have. Please see the attachment

This manuscript is a resubmission of an earlier submission. The following is a list of the peer review reports and author responses from that submission.
Round 1
Reviewer 1 Report
Identification and quantification of metabolomic biomarkers in patients with rheumatic diseases can be useful in disease diagnosis, monitoring activity and response to treatment, as well as assessing extra-articular symptoms. One of the extra-articular symptoms is skeletal muscle loss. The topic of metabolomic biomarkers in rheumatoid arthritis is not new, and many studies have been published, not only in animal models, but also in RA patients. Nevertheless, in my opinion, the aim of the study is interesting and current.
I have some comments for the authors.
We can expect that with the duration of the disease and increase in activity, loss of skeletal muscle also progresses. I suggest adding a comment regarding the variability over time, e.g. between day 55 and day 65 (Fig 6)
Verse 313
“Setting a cut-off point of 45 days for established muscle loss presence, we suggest the following list of biomarkers for muscle loss prognosis: 3-methylhistidine, 4-aminobutyric acid, acetylcholine, arginine, aspartate, carnosine, creatine, creatinine, glutamine, histamine, histidine, isoleucine, leucine, l-methionine, lysine, myo-inositol, n,n-dimethylglycine, n-acetyl alanine, n-acetylmethionine, pantothenate, phenylalanine, phosphocholine, phosphocreatine, pyridoxine, sarcosine, succinyl acetone and thiamine [25]. After 45 days, we suggest the following list as prospective biomarkers of muscle mass loss: 3-methylhistidine, 4-aminobutyric acid, acetylcholine, arginine, carnosine, creatine, creatinine, glutamine, histamine, histidine, isoleucine, leucine, l-methionine, lysine, myo-inositol, n,n-dimethylglycine, n-acetyl alanine, n-acetylmethionine, pantothenate, phosphocho line, phosphocreatine, pyridoxine, sarcosine, succinyl acetone and urocanate”
Repeated list. I suggest describing the difference, instead of repeating. The citation [25] is unclear. What suggestions do the authors have after 65 days?
In the discussion, I recommend distinguishing between biomarkers for disease activity or other and specific for muscle loss (for example, in a table). I suggest also defining the characteristic metabolic signature for muscle loss in RA.
In conclusion, I suggest adding a list of most specific biomarkers for the loss of skeletal muscles (as the title of the article suggests), according to the results of the study for RA patients.
Do the authors see any limitations in their research?
Reviewer 2 Report
In this work by Alabarse et al., urine of a mice model of RA with muscle loss is analysed for its metabolite composition, with the aim of identifying a biomarker signature in RA animals that can be used as a proxy for diagnosis/prognosis of muscle loss. The novelty of the work relies on the choice of urine as the biofluid for such analysis, which has clear advantages over plasma/serum in a clinical setting. Nonetheless, in my opinion several aspects need to be clarified before the manuscript is acceptable for publication, as follows:
- It should be made more clear throughout the manuscript that urine samples used for this study were collected from the same cohort of animals used in a previous study published by the same group (Alabarse et al. (2018), J Cachexia Sarcopenia Muscle, 9(3). It is particularly relevant to make this point more clear in the first section of results.
- Figures 1-5 should be provided with better image quality. Symbols in Figure 1 should be depicted at higher magnification.
- The authors state “As can be seen, these models [PCA and PLSDA] could segregate CO from CIA groups as well as CIA group in a time-dependent manner” (page 5, line 237). However, I believe PCA analysis in Figure 2 and Figure 4 do not show this segregation. A clear segregation between CIA and control groups, as well as between CIA animals at different timepoints of disease progression, can only be seen by PLSDA. Can the authors further discuss the validity and performance of both methods in the analysis of their data?
- Can the authors clarify what do they mean by “all the previously mentioned models from PLSDA provided sensitivity and specificity values to be considered, as shown in Table 1” (page 5, line 239)? It is not clear what are the “sensitivity and specificity values” the authors are referring to.
- In Table 1, can the authors clarify what are the differences between the datasets for CIA comparisons along time labelled as “BML” and “NMR”? What is the overlap of metabolites between both datasets? A Venn diagram could be included to illustrate this. What in these datasets could be triggering the identification of different major pathways implicated in a specific disease timepoint by the PLSDA analysis? For example, histidine pathway is identified with a 61% contribution to disease progression from day 55 to day 65 in the NMR dataset, but completely absent from the BML dataset: why does this occur?
- Please reformat Table 2 to better separate the data corresponding to each metabolite. In its current form it is not clear, for instance, if the VIP score of 2.8×10² (focused by the authors in page 11, line 292) corresponds to carnosine or aspartate.
- In page 14, line 313, the authors state “Setting a cut-off point of 45 days for established muscle loss presence”. Can the authors further justify the choice of this timepoint? It does not seem to match the timepoint of significant arthritis onset (day 35) reported for these animals in their previous work (Alabarse et al. (2018), J Cachexia Sarcopenia Muscle, 9(3)) neither the timepoint at which muscle morphometric parameters were analysed (day 65) in the same study. Even if this choice is based on the work published by the group in the study Filippin et al. (2013) J. Cachexia Sarcopenia Muscle, 4(3), the levels of muscle loss at this timepoint should be verified in the animals used in the current study.
- In page 14, can the authors further argument for the choice of the two sets of biomarkers of muscle loss proposed? It is not clear why all these metabolites were included as candidate biomarkers. I also suggest Figure 6 is complemented with a heatmap (or similar) showing the temporal variation of the levels of the relevant metabolites identified in the NMR analysis performed.
- Labels in Figure 7 are too small to be read. Figure needs to be reformatted or data should be presented in another format.
- In page 18, line 364, correct Table 2 to Table 3.